# Design and Performance Analysis of Foldable Solar Panel for Agrivoltaics System

**DOI:** 10.3390/s24041167

**Published:** 2024-02-10

**Authors:** Ramesh Kumar Lama, Heon Jeong

**Affiliations:** 1U Energy Co., Ltd., Annex Research Institute, 408, Building D5, Smart Park Knowledge Industry Ce 13 Gyoyuk-gil, Naju-si 58027, Republic of Korea; ramesh0527@uenergy.co.kr; 2Department of Fire Service Administration, Chodang University, Muan-gun 58530, Republic of Korea

**Keywords:** agrivoltaics system, foldable solar panel, renewable energy, solar tracking, sustainable agriculture

## Abstract

This study investigates the use of a foldable solar panel system equipped with a dynamic tracking algorithm for agrivoltaics system (AVS) applications. It aims to simultaneously meet the requirements for renewable energy and sustainable agriculture. The design focuses on improving solar energy capture while facilitating crop growth through adjustable shading. The results show that foldable panels, controlled by the tracking algorithm, significantly outperform fixed panels in energy efficiency, achieving up to a 15% gain in power generation and uniform power generation throughout the day. Despite the presence of shadows of adjacent panels in the early morning and late evening, the system’s effectiveness in creating microclimates for diverse crops demonstrates its substantial value. The foldable design not only protects crops from adverse climate conditions across different seasons but also generates energy efficiently. This demonstrates a step forward in sustainable land use and food security.

## 1. Introduction

The integration of photovoltaic (PV) systems into agricultural lands, known as agrivoltaics system (AVS), represents a promising convergence of renewable energy production with sustainable agriculture [1,2]. This dual-function approach not only increases power generation but also provides controlled shading to optimize crop growth conditions, contributing significantly to biodiversity protection and the efficient use of land resources [3]. The historical development of AVSs can be traced back to the foundational concept proposed by Adolf Goetzberger and Armin Zastrow in 1982 [4]. While the initial growth was gradual, recent years have witnessed a rapid expansion in AVS deployment. The installed capacity has seen an exponential increase, rising from approximately 5-megawatt peak (MWp) in 2012 to over 14-gigawatt peak (GWp) by 2021. This surge in adoption can be largely attributed to government funding initiatives in countries such as Japan (initiated in 2013), China (circa 2014), France (since 2017), the United States (beginning in 2018), and more recently, South Korea. These programs have been pivotal in advancing AVS technology and its global implementation [5]. Furthermore, the UN Climate Change Conference, COP26, has played a crucial role in reinforcing the importance of AVSs in the global renewable energy and sustainable agriculture agenda, by promoting international cooperation and policy support for such innovative solutions [6,7]. Several configurations and methods have been developed to optimize production in AVSs [8]. Since the AVS comprises power generation and agricultural production, the configuration can be defined based on the prioritization of outputs. The energy-prioritized approach emphasizes the optimization of solar energy generation, adapting agricultural practices to fit within the established framework of solar arrays, thereby cultivating crops between or beneath the panels [9]. In contrast, the agricultural prioritized approach places its primary focus on maximizing crop cultivation, with energy generation playing a secondary, however, integrated role, ensuring minimal disruption to traditional farming practices [8]. Bridging the gap between these two is the integrated agricultural and energy production approach, which seeks to design a system from the ground up that equally prioritizes both crop and energy production.

Different AVS methods implemented and tested so far have shown significant promise; however, like any emerging technology, it comes with challenges and limitations. In an energy-prioritized approach, focusing primarily on energy production can lead to suboptimal conditions for crop growth, especially if the solar panel setup interferes with the necessary sunlight, water, and nutrient requirements of the crops. A significant drawback is the inability to utilize 100% of the land area. This inefficiency stems from the complex nature of the installations for efficient solar tracking, which can leave portions of the land underutilized. Moreover, these systems often lack adaptive light control, meaning they cannot modulate the amount of shade provided to the crops beneath. This can lead to suboptimal growing conditions, as plants may either receive too much direct sunlight or be overly shaded. Another issue is the exposure of some areas to environmental extremes due to the design of the panels. For instance, crops can be susceptible to frost in the winter or excessive light and heat during the summer months. Furthermore, the uniformity in power distribution presents a problem. The power output of these systems tends to peak during midday when the sunlight is most direct, leading to an imbalance in energy production. This peak in power generation does not necessarily align with demand patterns, resulting in inefficiencies in energy use.

Collectively, these limitations highlight the need for advancements in AVS designs to enhance land use, adaptability to changing light conditions, protection against environmental extremes, and improved power distribution to better match consumption needs. Thus, we propose a foldable panel-based AVS. Figure 1 shows the conceptual illustration of the proposed AVS. The proposed method introduces a variety of features that offer substantial benefits both in agricultural productivity and power generation efficiency.

In terms of agricultural efficiency, the foldable modules provide adjustable light with tilt angles ranging from 10 to 89 degrees. This flexibility in positioning allows for precise control over the amount of sunlight reaching the crops, optimizing photosynthesis throughout various growth stages. During winter, the ability to position panels at a shallow angle of 0–10 degrees can offer frost protection by disrupting cold air flow at dawn, a critical time for frost formation. In the summer, the system can be adjusted to allow only 20 to 50% of light to reach the crops, thereby preventing damage from excess light and heat stress. Additionally, the height adjustability of the system is a crucial feature, facilitating the use of large machinery for planting, care, and harvesting without the need for system dismantlement or complex maneuvering.

The benefits extend to power generation as well. The solar tracking feature ensures that more power is generated compared to fixed-panel systems, as it allows for the proper orientation of panels to capture the maximum amount of sunlight throughout the day. This leads to uniform power generation; with a dual-panel-based method, east-facing panels generate uniform power before noon while west-facing panels take over in the afternoon, smoothing out the typical midday peak in power production and aligning better with energy consumption patterns.

Simplicity is another feature of the proposed system. The system is characterized by a modular design consisting of an array of foldable panels. This design facilitates effortless scalability; expanding the system is as simple as adding more panels while reducing its size only requires the removal of existing modules.

## 2. Related Works

Since its inception, the development of AVSs has been shaped by both industrial implementations and academic research. Projects [10,11,12,13,14] demonstrate the participation of various industrial institutes in the implementation of AVSs. These projects encompass the use of both fixed panels for agriculture-centric solutions and tracking-based systems, including single-axis and dual-axis designs, for energy-centric AVS applications. Notably, Next2sun’s technology integrates fixed vertical installations with bifacial PV modules oriented east and west, creating approximately 10-meter gaps between rows for agricultural activities [10]. Agrinergie^®^, developed by Akou Energy, features two strategically spaced module strips, allowing for the cultivation of lemongrass in the intervening spaces [11]. A significant example is the Agrovoltaico^®^ System, developed by REM Tec and the University of Piacenza in Italy [12,13], which includes a solar tracking mechanism tailored to maximize energy production in agrivoltaics settings. This patented technology represents a substantial leap forward in improving energy efficiency in agrivoltaics installations. Furthermore, BayWa r.e.’s installation in the Netherlands showcases a large-scale approach to integrating agrivoltaics into fruit production [14]. This project stands out as a prime example of harmonizing energy production with fruit farming in AVSs, making it a significant case study within the European context.

Similarly, various academic researchers have conducted studies in the field of AVSs. A study [15] employs single-axis tracking schemes for bifacial solar panels. This aims to balance the sunlight distribution between photovoltaic arrays and crops, tailored to the plants’ photosynthetic needs. The system dynamically adjusts between standard and reverse sun tracking throughout the day and across seasons, optimizing the capture of solar energy while meeting crop light requirements. A simulation study of a patented AVS named ‘Agrovoltaico^®^’, where maize crops were grown under solar panels in northern Italy, revealed that panel density had a larger impact on reducing radiation than the sun-tracking setup did. The dual-axis tracking of solar positions is implemented in work [13]. Utilizing a dual-axis tracking mechanism and artificial intelligence (AI) techniques [16], the system claims to improve power generation by up to 52%, offering better crop management through weather monitoring and data analysis. A study [17] examines the potential of AVSs on grape farms in India, showing that incorporating solar PV panels can significantly increase farm revenue while maintaining grape yields and potentially offering substantial national electricity generation. A study [18] investigates the impact of solar panels on the microclimate and biomass productivity of an unirrigated pasture in Oregon, finding that solar panels result in higher soil moisture, late-season biomass, and improved water efficiency. A study [19] modeled a 20 MW solar PV plant across various locations in South India using the System Advisor Model to compare fixed, single-axis, and double-axis solar tracking systems. It found that the double-axis tracking system was the most efficient and economically viable, yielding the highest annual energy generation and capacity factor, with positive net present values. In contrast, fixed and single-axis systems were economically unfeasible. Hyderabad was identified as the optimal location for this solar PV project based on the lowest leveled cost of energy and the best project feasibility. With the recent development of AI technologies, a data-driven approach in AVSs has also been introduced [20,21,22,23,24]. Study [20] showcases a data-driven approach in developing an optimization model for a vertically mounted AVS, where empirical data underpin the integration of solar radiation, photovoltaics, and crop yield sub-models. A multidisciplinary framework for urban rooftop AVSs in Shenzhen combines GIS, biogeochemical, and solar power simulations, emphasizing a data-centric assessment for food and energy production [21]. Study [22,23] proposes an AVS architecture in the U.S., utilizing data-intensive tools like micrometeorological analyses and irradiance modeling software, coupled with detailed crop data, to optimize energy and agricultural output. Research [24] underscores the use of the Internet of Things (IoT) and big data in AVSs, focusing on the technical integration of agriculture and photovoltaics, and highlighting how data analytics can facilitate balanced land use strategies for simultaneous crop and energy production. Additionally, numerous studies have been conducted on the integration of AVSs with greenhouse tunnel farming [25]. While a standalone AVS offers significant benefits, its integration with greenhouse tunnel farming amplifies these advantages through improved microclimate control, the direct and efficient use of generated energy, and potential structural and economic efficiencies [26,27,28,29,30].

## 3. Proposed Agrivoltaics System

We propose a novel AVS model that utilizes foldable solar panels for energy generation. This model is coupled with a single-axis tracking method to follow the sun’s position, thereby enabling efficient power generation. We divide the proposed system into two parts: the mechanical design and the solar tracking mechanism. The mechanical design is characterized by a foldable module design, while the solar tracking is defined by an algorithm that tracks the position of the sun. Initially, our system utilizes a time-based single-axis method [31] called Algorithm 1. Given that our system incorporates an array of foldable panels, the conventional time-based single-axis tracking is particularly susceptible to the shadow effect in the early morning and late evening. To counteract this, we implement a shadow-controlled improved time-based algorithm, which we designate as Algorithm 2, aimed at minimizing this shadow effect.

### 3.1. Foldable Module Design

We design the foldable module using a dual-panel structure, where one edge of each panel is joined to its counterpart, while the opposite edge is connected to a movable base. This innovative configuration facilitates the adjustment of the panel’s tilt angle to enhance solar energy capture, as depicted in Figure 2. The east-facing panel is designed to harness solar energy from sunrise to midday, while the west-facing counterpart captures the sun’s rays from midday to sunset. The slope of the panels can be adjusted by changing the height of the pivot point. This allows the panels to track the sun’s movement across the sky, which helps in appropriate panel exposure to sunlight throughout the day. The three-dimensional representation in Figure 3 illustrates the folder module’s spatial orientation. For practical AVS implementation, multiple modules are arrayed in series and oriented towards the east and west to maximize solar energy throughout the day.

The three-dimensional representation in Figure 3 illustrates the folder module’s spatial orientation. For practical AVS implementation, multiple modules are arrayed in series and oriented towards the east and west. To maximize solar energy, the tilt angle of the panel is adjusted according to the position of the sun throughout the day.

For ease of analysis, we depict only the east-facing folder module in Figure 3 as a flat plane as shown in Figure 4. For the efficient collection of energy, the normal vector of the panel should be parallel to the direction of incoming rays from the sun. This leads to the design of the folder module as a plane rotating around a horizontal axis and forming the tilt angle β, as shown in Figure 4.

For a power generation site identified by its latitude and longitude, and considering the specific day of the year, the established procedure sequentially calculates the position of the sun at each time interval. This yields the solar azimuth $\psi_{s}$ and altitude $\gamma_{s}$ as outlined in [32]. The sun’s position can be represented as a vector pointing from the panel to the sun, which can be calculated as
(1)$$\begin{array}{l} x=\cos\gamma_{s}\sin\psi_{s}\\ y=\cos\gamma_{s}\cos\psi_{s}\\ z=\sin\gamma_{s} \end{array}$$
where we assume the azimuth angle $\psi_{s}$ is measured from the north; thus, $\psi_{s}$ = 0 would mean the sun is due north and $\psi_{s}$ = 90° would be east. The tilt angle β and vector $\vec{n}$ can be mathematically defined using Cartesian coordinates for precise alignment with the sun’s position.
(2)$$\begin{array}{c} n_{x}=\sin\beta \sin\psi_{p}\\ n_{y}=\sin\beta \cos\psi_{p}\\ n_{z}=\cos\beta \end{array}$$The y-component is 0, indicating no tilt towards the south or north and perfectly oriented in the east and west direction. In this case, the azimuth angle of the panel is perfectly perpendicular to the north direction (Y-axis) with $\psi_{p}=90{}^{\circ}$. The normal vector can be expressed as
(3)$$\vec{n}=\left[\sin\beta ,0,\cos\beta \right]$$Figure 5 shows the bi-directional solar panel system designed to collect solar irradiance from both east and west directions. This system is depicted on a two-dimensional plane where the vertical and horizontal axes demarcate the panels’ orientation. Vectors ${\vec{n}}_{east}$ and ${\vec{n}}_{west}$ emerge from the surface planes facing east and west, respectively. The normal vector ${\vec{n}}_{east}$ with a tilt angle $\beta_{1}$ is expressed as
(4)$${\vec{n}}_{east}=\left[\sin\beta_{1},0,\cos\beta_{1}\right]$$Similarly, the normal vector ${\vec{n}}_{west}$ with a tilt angle $\beta_{2}$ is given by
(5)$${\vec{n}}_{west}=\left[\sin\beta_{2},0,\cos\beta_{2}\right]$$

### 3.2. Time-Based Single-Axis Tracking Algorithm

We employ the conventional time-based single-axis solar tracking method to monitor the sun’s trajectory, aiming to maintain the solar panel’s normal vector parallel to the solar light’s incident angle. This alignment is pivotal for maximizing sunlight exposure throughout the day, thereby enhancing the energy capture efficiency of the solar panels. In this algorithm, the input is the time value t of any given moment, and the output is the tilt angle β1(t) and β2(t) of the panel. By continuously adjusting the panel angles about the sun’s position, the algorithm ensures the appropriate orientation for energy generation. It adopts a straightforward yet effective approach to linearly adjust the panel’s tilt angle within preset maximum and minimum values, corresponding with the time of day to emulate the sun’s apparent motion. At sunrise, the panel is set to its maximum tilt angle of 89 degrees. As midday approaches, β1(t) it progressively decreases linearly. The angle reaches its minimum of 10 degrees at noon. After midday, β2(t) is incrementally restored until sunset, when the panel reverts to the maximum angle.
**Algorithm 1: Time-based Solar Tracking****Definitions**tsr=Sunrise time.tss=Sunset time.β1(t)=Solar panel tilt angle of east facing panel at time tβ2(t)=Solar panel tilt angle of west facing panel at time tβmax=Maximum tilt angle, which is 89°βmin=Minimum tilt angle, which is 10°Δt:Solar day duration, time duration from tsr to tssΔt1:Time duration from given time t to sunrise time tsrΔt2:Time duration from given time t to solar noon timeneast(t)=Normal vector repersenting east facing panels orientation at time tnwest(t)=Normal vector repersenting west facing panels orientation at time t**Angle control**For tsr≤t<solar noon  β1(t)=βmax−(βmax−βminΔt/2)×Δt1  neast(t)=[sinβ1(t),0,cosβ1(t)]For solar noon≤t≤tss  β2(t)=βmin+(βmax−βminΔt/2)×Δt2  nwest(t)=[sinβ2(t),0,cosβ2(t)]

### 3.3. Shadow Effect

In the early morning and late afternoon, partial shading between two adjacent panels occurs, resulting in a reduced energy harvest, particularly during this period. Figure 6a shows the occurrence of shadows in the neighboring panel cast by the front panel [33,34]. The length of the shadow as shown in Figure 6b from the center of the front panel is represented by *S_L_* and is estimated as
(6)$$S_{L}=P_{L}\times \frac{\sin\beta }{\tan\gamma_{s}}$$To reduce the length of shadow below *S_L_* compared to the distance between panels *D_L_*, one option is to widen the spacing between panels, which, although effective in reducing shadowing, consequently requires a larger installation space.

Additionally, this approach comes with an increase in installation costs. The second solution is to equip each panel with an independent tilt mechanism, using dedicated actuators to finely adjust angles and prevent shadowing on adjacent panels. This would ensure maximum solar incidence on each panel, and improved energy production. However, this increases the complexity and cost of the system due to the additional motors and their associated energy consumption. To increase energy production and ensure cost efficiency, we adjust the solar panel tilt angles at critical times to minimize shadow effects, as described in Algorithm 2. We empirically determine the proper tilt angle to achieve maximum power output at specific times. The power output is measured using a WT3000 power analyzer, and the angle is adjusted manually at these crucial times.
**Algorithm 2: Improved Time-based Solar Tracking****Definitions**tsr=Sunrise time.tmm=Mid morning time.tprn=Pre noon time.tpon=Post noon time.tln=Late noon time.tss=Sunset time.β1(t)=Solar panel tilt angle of east facing panel at time tβ2(t)=Solar panel tilt angle of west facing panel at time tβsr=Tilt angle at tsr is set 7.3°βmm=Tilt angle at tmm is set as 30.4°βprn=Tilt angle at prenoon tprn is set as 1°βpon=Tilt angle at post noon tpon is set as 1°βln=Tilt angle at late noon tln is set as 41°βss=Tilt angle at tss sunset is set as 18.8°Δt1:Time duration from time t to sunrise time tsrΔt2:Time duration from time t to mid morning time tmmΔt3:Time duration from time t to post noon time tponΔt4:Time duration from time t to late noon time tlnneast(t)=Normal vector repersenting east facing panels orientation at time tnwest(t)=Normal vector repersenting west facing panels orientation at time t**Control Algorithm**:**Angle control**For tsr≤t<tmm  β1(t)=βsr+(βmm−βsrtmm−tsr)×Δt1  neast(t)=[sinβ1(t),0,cosβ1(t)]For tmm≤t<tprn  β1(t)=βmm−(βmm−βprntprn−tmm)×Δt2  neast(t)=[sinβ1(t),0,cosβ1(t)]For tprn≤t<tponIf t≤tprn+tprn2  β1(t)=βprn  neast(t)=[sinβ1(t),0,cosβ1(t)]Else: β2(t)=βprn  nwest(t)=[sinβ2(t),0,cosβ2(t)]For tpon≤t<tln  β2(t)=βpon+(βln−βpontln−tpon)×Δt3  nwest(t)=[sinβ2(t),0,cosβ2(t)]For tln≤t<tss  β2(t)=βln−(βln−βsstss−tln)×Δt4  nwest(t)=[sinβ2(t),0,cosβ2(t)]

## 4. Experiments

This section aims to validate the power generation efficiency of the foldable-panel-based system designed for the AVS. For that first, we set up the physical infrastructure to collect power and associated data. Next, we implemented the time-based single-axis tracking to analyze power-generated data and analyze the shadow effect. Next, we implemented the improved time-based shadow-control algorithm. We collected the power generation from foldable panels and fixed panels. For that, we installed the physical infrastructure to collect the solar power and related data. The physical infrastructure consisted of mechanical and electrical components. Mechanical components consisted of mounting structures. These are the supports that held the solar panels in place. They included rails, brackets, and a foundation or poles that secure the entire system to the ground. The sliding base is the horizontal bar that allows one end of the panels to move linearly, resulting in a change in the slope of the panel. Motors moved the solar panels to move on the sliding base. Similarly, electrical components consisted of PV modules, controllers for the solar tracker, and inverters. Initially, we installed and labeled the solar panels and inverters, along with irradiation sensors. A total of eight panels were installed and arranged into four pairs. Pairs a and b comprised foldable panels, which together constituted the folder modules outlined in the preceding section. To facilitate performance comparison, we installed fixed panels in pairs c and d. For uniform power collection, the east-facing panels from pairs a and b were connected to the inverter labeled ‘Inverter Movable East’ (IME), while the west-facing panels from these pairs were connected to the ‘Inverter Movable West’ (IMW). Similarly, the east- and west-facing panels from pairs c and d were linked to the ‘Inverter Fixed East’ (IFE) and the ‘Inverter Fixed West’ (IFW), respectively. This led to the load balancing to the grid. The energy demand often peaks in the morning and late afternoon or early evening. By having separate inverters, the system can more effectively balance the load, providing power in alignment with demand patterns. Four irradiation sensors were positioned, one for each pair of panels, to measure solar irradiance accurately. Additionally, closed-circuit television (CCTV) cameras were installed to monitor weather conditions and cloud coverage. Data from these instruments were aggregated and logged through a remote terminal unit (RTU), as depicted in Figure 7.

### 4.1. Data Collection

Our physical infrastructure for solar energy experimentation was set up in Suncheon City, South Korea. The experiments were conducted from the 15 to the 25 November 2023, under varying weather conditions, ranging from clear skies to cloudy conditions. Sunrise and sunset times were sourced from the Korean government’s weather and meteorological forecasting division. Notably, on a representative day, sunrise was at 07:04, solar noon at 12:14, and sunset at 17:25, marking a total daylight duration of 10 hours and 21 min. Specific data points include sunrise at 7:10 and sunset at 17:23 on the 21st, and sunrise at 7:13 with sunset at 17:22 on the 25 November. The solar panels employed in our study are capable of handling solar irradiation ranging from 0 to 1000 W/m^2^. Their electrical characteristics, as specified by the manufacturer, include a current range of 0 to 12 Amperes and a voltage range of 0 to 54 Volts, varying according to the load resistance.

Each panel measures 1038 mm in width, 2228 mm in length, and 35 mm in thickness, with a module conversion efficiency of 21.2%. For energy conversion, we used a single-phase inverter with a maximum power rating of 3.5 kW. This inverter operates within a voltage range of 100–500 V and has a Maximum Power Point Tracking (MPPT) range of 260–400 V. Power data were collected from the inverter in kW units at 10-minute intervals. Additionally, solar irradiation data were measured using a PYR20 Pyranometer sensor, capable of measuring up to 2000 W/m^2^ within a spectral range of 400–1100 nm.

### 4.2. Performance Evaluation

We evaluated the performance of the proposed method using two metrics, percentage gain, and peak variance ratio, with respect to fixed-panel-based modules. The percentage gain (*PG*) in daily energy production was calculated using the standard formula:(7)$$PG=\frac{CPMo-CPFi}{CPFi}\times 100\%$$
where CPMo and CPFi are the cumulative power generated by foldable modules and fixed modules, and estimated as
(8)$$CPMo=\sum P_{IME}\left(t\right)+\sum P_{IMW}\left(t\right)$$
$$CPFi=\sum P_{IFE}\left(t\right)+\sum P_{IFW}\left(t\right)$$
where $P_{IME}$ and $P_{IMW}$ are the power generated by inverters connected to foldable panels facing the east and west directions. $P_{IFE}$ and $P_{IFW}$ are the power generated by inverters connected to fixed panels facing the east and west directions. Uniform energy generation throughout the day ensures the receipt of abundant sunlight at different times of the day. Thus, the risk of a substantial drop in energy generation due to transient weather conditions is mitigated. This approach can be particularly beneficial in regions where cloud cover is unpredictable and frequent. The increased amount of energy generation was estimated by comparing the cumulative power generation of folder modules with the power generation of fixed modules. We used the peak variance ratio to determine whether the foldable panels generate uniform power or not as compared to fixed panels. To calculate it, we estimated the peak values of foldable and fixed panels first.
(9)$$PPINF\left(t\right)=\max\left(P_{IFE}\left(t\right),P_{IFW}\left(t\right)\right)$$
(10)$$PPINM\left(t\right)=\max\left(P_{IME}\left(t\right),P_{IMW}\left(t\right)\right)$$Next, we estimated the standard deviation of peak values of PPINF and PPINM as *STD_PPINF_* and *STD_PPINM_*. Finally, the peak variance ratio was estimated as
(11)$$STD_{PP}=\frac{STD_{PPINM}}{STD_{PPINF}}$$Here, if the ratio is close to 1, it suggests that the peak variances are similar. If the ratio is much greater than 1, it suggests that the ‘Peak Values Movable’ data have a higher variance than the ‘Peak Values Fixed’ data. Conversely, if the ratio is much less than 1, it suggests that the ‘Peak Values Movable’ data have a lower variance than the ‘Peak Values Fixed’ data. A higher variance in the power output of the panel suggests that there is a wider range of power output values throughout the day. This indicates that the panel is generating power over a broader range of times, suggesting the power being spread from morning to evening. The peak power may not be as high as that of the fixed panel at any given moment, but it is more consistent across different times. A lower variance in the power output of the panel indicates that the values are more clustered together, which suggests that there is a shorter period during the day when the power output is near its peak. This occurs when the sun is at the best angle for the fixed panel, resulting in a concentrated time of high-power output.

### 4.3. Results

We conducted tests on 15 November 2023, using a time-based algorithm, Algorithm 1, under ideal conditions of a fully sunny day with zero cloud presence in the sky. Sunrise occurred at 07:04, noon was at 12:14, and sunset was at 17:27. Figure 8a shows the power generation plots of all panels. For clearer visibility, we present the same generation data separately in Figure 8b,c. Figure 8b displays the power generation pattern of the foldable panels facing east and west, while Figure 8c shows that of the fixed panels facing in these directions. Figure 8d presents the total power generated by both foldable and fixed panels. This was mathematically calculated as the summation of the power generated by the east-facing and west-facing fixed and foldable panels. Examining the power generation pattern, we observe that power is not being generated in the early morning as expected. This is due to the use of Algorithm 1 to control the panel slope angle, which does not consider the shading effect. Consequently, the shadow of the foremost panel is cast directly onto the adjacent panel, affecting power generation during the early morning. The shadow of the front panel cast on an adjacent panel at the test site is shown in Figure 9. The percentage gain in power generation on this day is negative, as shown in Table 1.

Figure 10 displays the peak value distribution of the power generated on the same day. This figure reveals that the power data generated by the fixed panels are clustered around midday, in contrast to the power data from the foldable panels. The peak variance improvement ratio, at 1.11, is greater than 1, indicating a more uniform power generation throughout the day in foldable panels.

On 21 November 2023, we conducted additional tests, this time employing Algorithm 2 which tracks solar position considering the shadow cast on adjacent panels. As described in the previous section, we adjusted the tilt angles of the panels at crucial moments to improve solar exposure and reduce the shaded area on the second panel. The weather conditions were similar to those of the previous test, with sunrise at 7:10 and sunset at 17:23. Figure placement is similar to Figure 8. Figure 11a illustrates the power generation patterns of the respective panels. From Figure 11b, it is visible that the power generated by the foldable east-facing panel is improved compared to Figure 8 at the same time. We can attribute this gain to Algorithm 2. Figure 11d compares the total power generated by foldable panels against fixed panels. This graph clearly shows that foldable panels generated more power than fixed panels most of the time. Table 1 indicates a 4.66% gain from this setup. Figure 11b illustrates the power generation pattern of the foldable panels, while Figure 11c depicts the pattern for the fixed panels. Similarly, Figure 12 confirms the uniform power production, with a peak variance ratio of 1.13.

Algorithm 1 refers to Algorithm 1 derived in Section 3.2Algorithm 2 refers to Algorithm 2 derived in Section 3.3P_IFE_: Cumulative sum of the power of inverter connected to east-facing fixed panel in kilowatts.P_IFW_: Cumulative sum of the power of inverter connected to west-facing fixed panel in kilowatts.P_IME_: Cumulative sum of the power of inverter connected to east-facing foldable panel in kilowatts.P_IMW_: Cumulative sum of the power of inverter connected to west-facing foldable panel in kilowatts.

Similarly, tests were conducted on 25 November 2023, using Algorithm 2 under identical weather conditions with sunrise at 7:13 and sunset at 17:22. Figure 13 and Figure 14 show the power generation pattern and the peak value plots. The power generation graph follows the same pattern as in the previous test environment employing Algorithm 2. Table 1 shows that foldable panels achieved a noteworthy power gain of 8.14% compared to fixed panels, with a peak variance improvement ratio of 1.12 on this day.

Figure 15 illustrates the comparison of cumulative power generated by foldable panels and fixed panels oriented in the east and west directions at different. It is evident from the figure that, on most days, foldable panels generate more power than their fixed counterparts. More specifically, east-facing foldable panels produced less power on November 15th and 20th. This reduced power generation can be attributed to the use of a time-based algorithm designed to track the sun’s movement, which does not account for the shadow from adjacent panels. Consequently, power generation in the early morning is significantly impacted by shadows cast by the foremost panel. However, this effect is not as pronounced in west-facing panels.

Figure 16 displays the power generation gains of the proposed foldable model. The graph clearly shows that starting from 21 November, the foldable model has achieved a substantial increase in power generation, indicating a significant enhancement in solar energy capture. This improvement demonstrates the efficiency of the system’s design configuration, which is largely due to the ability of the design to dynamically adjust the panel orientation. This ensures proper solar alignment throughout the day and is a testament to the meticulous consideration given to panel placement, aimed at reducing shading and maximizing solar exposure. The precision of the algorithm is further evidenced by the marked improvements in energy generation on specific dates. Notably, on 25 November and 8 December, the power generation gains for both east- and west-facing panels reached their zenith. These increments in gain percentage are the result of the algorithm’s accurate calculations and its capacity for real-time adjustments, tailored to match the patterns of solar irradiation with minimal margin for error. The consistently positive gains observed on these dates, and particularly the highest gains, affirm the algorithm’s efficacy in mitigating the shadow effect and enhancing energy harvest throughout the various sunlight periods.

Additionally, to validate the power generated by the solar panels and the corresponding alternating current by inverters, we collected the irradiation data of individual solar panels, simultaneously with power data. Figure 17 shows the comparison of power generation data and the irradiation data of the corresponding panel. Graphically, we can see that the power production pattern and irradiation pattern resemble each other in all panels. We scaled down the irradiation value by a scalar quantity of 500 to match the value of power. We estimated the correlation between power and irradiation data. Figure 18 shows the corresponding correlations of power and irradiation of the panels. From the correlation graph, we can see that the power generation of the panels and the irradiation on those panels are highly correlated. The highest correlation was 0.98 in fixed east panels and the lowest was 0.94 in movable east panels. This graph validates that the power generated by inverters is genuinely generated from solar power without external interference. The power generation curve would deviate noticeably from the solar irradiation curve if there were significant losses due to system inefficiencies like inverter losses, resistance in cables, and faults in the system. Thus, the close resemblance indicates that such losses or faults are minimal or non-existent. A close correlation of the power and irradiation data suggests that the solar panel system is efficiently converting solar irradiation into electrical power. This is an indicator of good system health and efficient performance. The similarity in the patterns of both plots implies that the solar panels are functioning as expected. They are effectively responding to the variations in solar irradiation throughout the day or over different days. It also validates the accuracy of the solar irradiation measurements.

## 5. Discussion

Our study presents a comprehensive evaluation of the power generation capabilities of foldable versus fixed solar panels, using time-based tracking algorithms under optimal conditions. The data collected on various dates provide clear insights into the effectiveness of the algorithms and panel designs in maximizing solar energy capture. On 15 November 2023, the implementation of Algorithm 1 under ideal sunny conditions resulted in power generation from early morning, as expected. However, the shading effect due to the panel’s orientation led to a negative gain in power generation. This was particularly evident in the east-facing foldable panels, where the shadow cast by the foremost panel significantly reduced early morning power generation. Despite the ideal conditions, the lack of consideration for adjacent shadows in Algorithm 1 was a limiting factor, highlighting the critical importance of accounting for panel placement and orientation in the design of tracking algorithms. Subsequent tests on 21 November 2013 and 25 November 2023, using Algorithm 2, which considered the shading fraction, showed marked improvements. The power generation patterns for these dates demonstrate the superiority of Algorithm 2 over Algorithm 1. By adjusting tilt angles to improve solar exposure, foldable panels consistently outperformed fixed panels, as evidenced by the 4.66% and 8.14% gains in power generation, respectively. These results underscore the potential of dynamic panel orientation to enhance solar energy capture, especially in foldable panel designs. The cumulative power generation data gain of up to 15% (Figure 16) further validates the effectiveness of foldable panels. Despite the reduced power generation on 15 November and 20 November due to Algorithm 1’s limitations, subsequent dates showed increased power output from the foldable panels, with significant gains starting from 21 November. This aligns with the implementation of Algorithm 2 and signals a clear turning point in the effectiveness of our solar panel system.

The design of the foldable solar panel system offers distinct advantages for agriculture. The system’s reduced structural complexity—requiring fewer poles—minimizes obstructions in the field, allowing for efficient land use and minimal disturbance to agricultural activities. Even though this study does not include the test data related to it, this scenario can be subjectively evaluated. The adjustable tilt angles and the height-based design provide adaptive shading for crops and accommodate the use of large agricultural machinery. This ensures that farming operations proceed unhindered by the solar infrastructure. The empirical data collected support these design benefits, showing that the implementation of the solar tracking algorithm does not compromise crop growth conditions. The dynamic adjustment of the panels based on the solar tracking algorithm allows for precise control over the amount of sunlight reaching the crops beneath. By maximizing the variance in peak voltage, the system can ensure that the panels are positioned to capture the maximum amount of sunlight when necessary and adjust to provide adequate shade to the crops when too much direct sunlight could be harmful. This balance is crucial for maintaining optimal photosynthetic activity and protecting plants from stress caused by excess light and heat. This result validates the second claim of enhanced agricultural production.

**Environmental and economic considerations:** The environmental analysis of AVS based on the implementation [35] indicates the potential for significant land use efficiency and carbon footprint reduction, given the dual functionality of energy generation and agriculture. The creation of favorable microclimates for diverse crops and the conservation of soil and water resources reinforce the environmental sustainability of this approach [36]. Economically, foldable solar panel systems demonstrate a promising trajectory. While initial costs and potential maintenance present considerations for investment, the long-term benefits in energy savings and possible crop yield improvements suggest a favorable return on investment. Future studies should incorporate a detailed cost–benefit analysis, including long-term durability assessments, to fully understand the economic implications. Furthermore, scalability and market adoption are key to realizing the economic benefits of AVSs. With advancements in technology and increased adoption, economies of scale may reduce costs, making the foldable solar panel system more accessible and economically viable [37]. Incentives and subsidies could also play a crucial role in accelerating the adoption of such sustainable practices.

**Limitations:** While our findings reveal the benefits of employing dynamic tracking algorithms and foldable panel systems for enhanced solar energy capture, our tests were confined to specific conditions and algorithms. Although the primary application of this model is intended for AVSs, the current study does not include test data related to crop yield and other relevant agronomic parameters. Furthermore, a comprehensive cost–benefit analysis remains to be conducted. These aspects are critical for a full evaluation of the system’s practicality and economic viability.

## 6. Conclusions

This study has investigated the effectiveness of a foldable solar panel system equipped with a dynamic tracking algorithm in the context of the Agrivoltaics System (AVS). Our principal aim was to address the simultaneous demands for renewable energy generation and sustainable agriculture. The innovative design emphasizes efficient solar energy capture while also facilitating crop cultivation through adjustable shading. Our results decisively show that foldable panels controlled by the tracking algorithm are superior to fixed panels in energy efficiency, achieving up to a 15% improvement in power generation. Moreover, these panels deliver consistent power output across the day. Despite encountering challenges with shaded areas during the early morning and late evening, the system has proven exceptionally effective in creating favorable microclimates for diverse crop types, highlighting its considerable promise. The foldable panel system not only safeguards crops from the unpredictability of weather throughout the seasons but also guarantees efficient energy production. As far as our knowledge goes, this is the first instance of such an implementation in AVS applications. Although this study does not encompass data on crop yield and other related agronomic parameters, it establishes a solid foundation for future research into foldable solar panels in sustainable AVSs.

**Future directions:** The control algorithm could be refined to accurately track the sun’s position, considering specific geographic locations, the day of the year, and the time of day. The tilt angle of the panels might be dynamically adjusted, especially to mitigate the adjacent shadow that significantly affects power generation in the early morning and late afternoon. Additionally, simulations of agricultural yield based on controlled shading rates under varying weather conditions could be implemented. This would provide a more holistic understanding of the system’s efficiency and its impact on agricultural productivity.

## Figures and Tables

**Figure 1 sensors-24-01167-f001:**
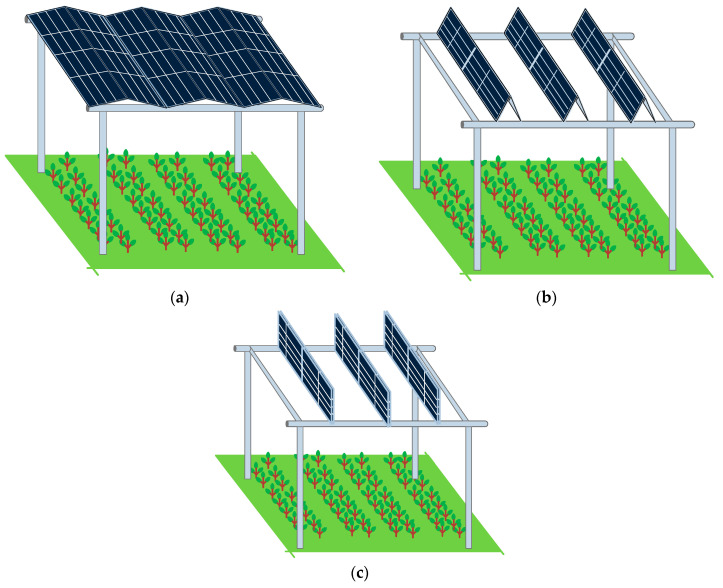
Conceptual illustration of a foldable-panel-based AVS. (**a**) Complete shadow. (**b**) Partial shadow. (**c**) Complete open.

**Figure 2 sensors-24-01167-f002:**
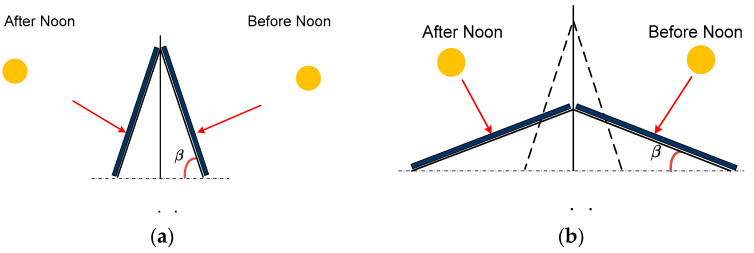
Energy collection. (**a**) High tilt angle. (**b**) Low tilt angle.

**Figure 3 sensors-24-01167-f003:**
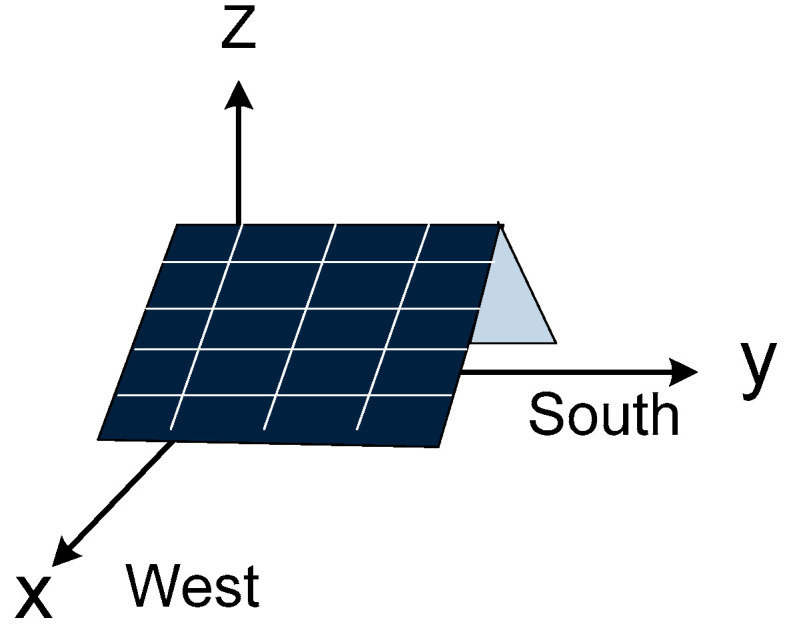
Three-dimensional view.

**Figure 4 sensors-24-01167-f004:**
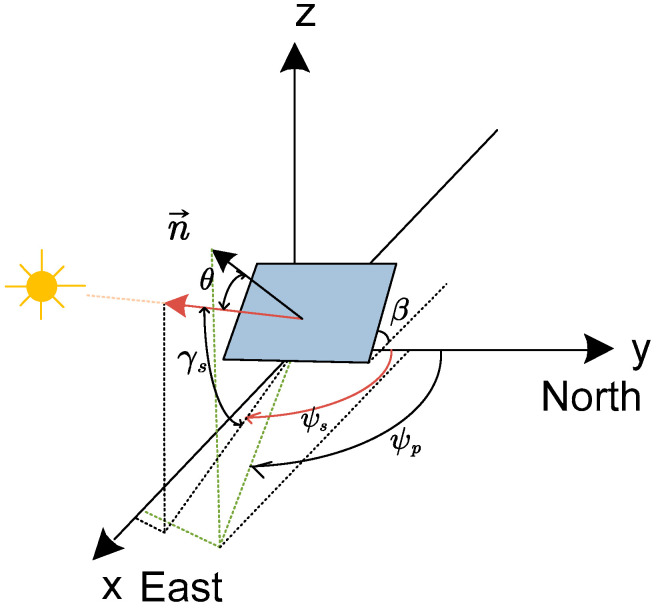
Vector geometry of single-axis east-facing foldable panel.

**Figure 5 sensors-24-01167-f005:**
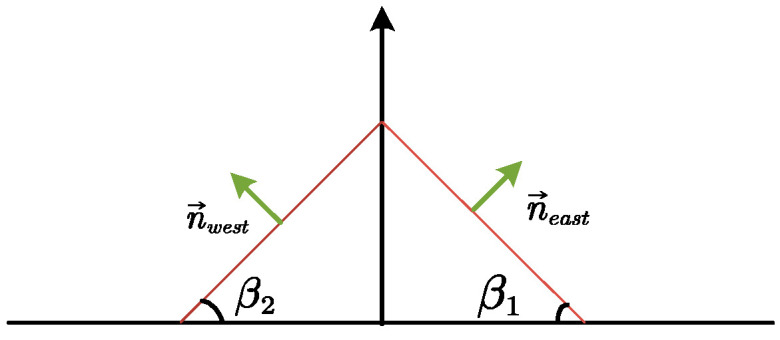
Vector geometry of single-axis dual-sided solar panel.

**Figure 6 sensors-24-01167-f006:**
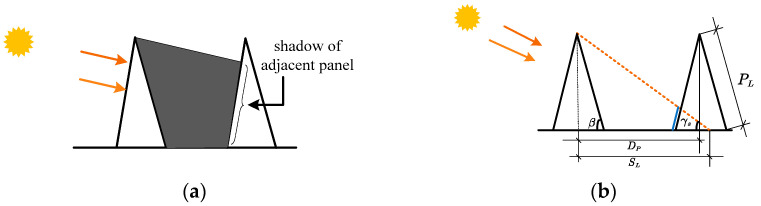
Shadow effect. (**a**) Shadow occurrence. (**b**) Shadow length estimation.

**Figure 7 sensors-24-01167-f007:**
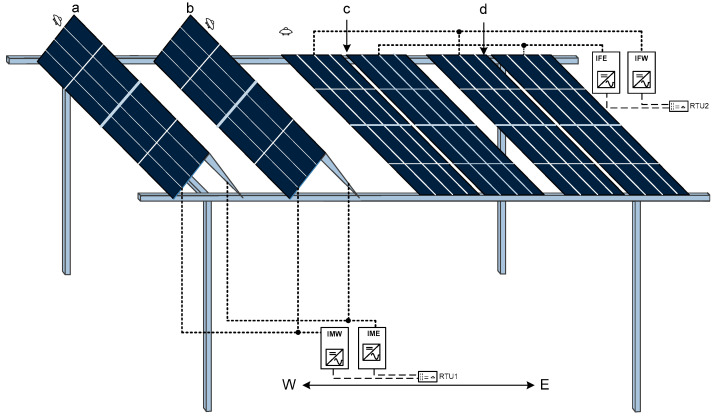
Panel installation configurations. (a,b) Foldable panel pairs (c,d), Fixed panel pairs.

**Figure 8 sensors-24-01167-f008:**
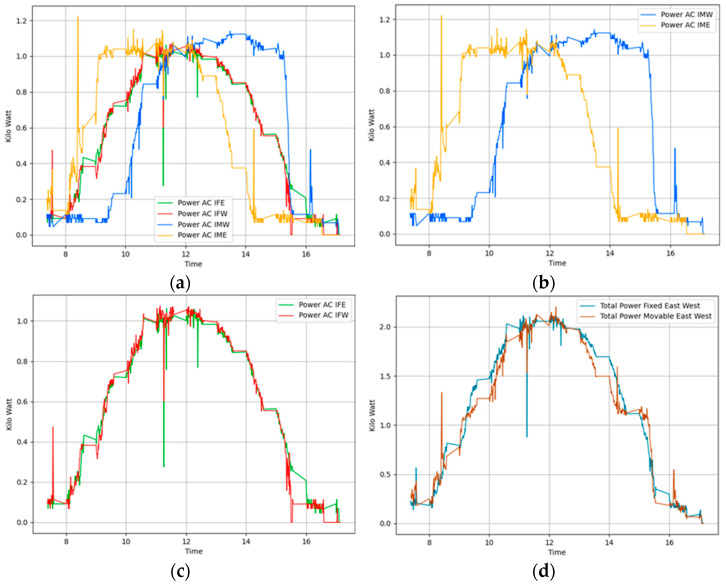
Power generation plots based on solar tracking Algorithm 1 Day1 (**a**) All panels. (**b**) Movable east-facing and movable west-facing panels. (**c**) Fixed east-facing and fixed west-facing panels. (**d**) Total power generated by movable and total power generated by fixed panels.

**Figure 9 sensors-24-01167-f009:**
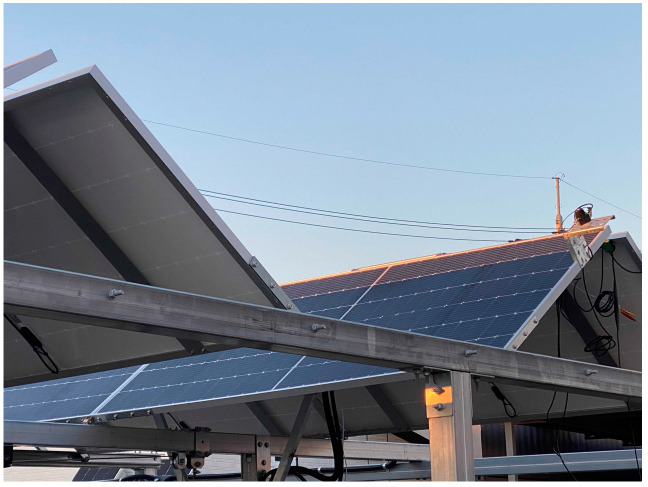
The shadow was cast on an adjacent panel at 7:24.

**Figure 10 sensors-24-01167-f010:**
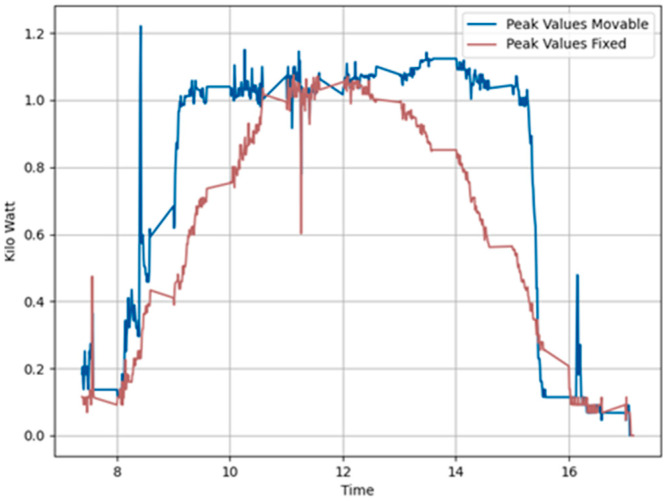
Peak variance ratio of Day 1 (20231115) Algorithm 1.

**Figure 11 sensors-24-01167-f011:**
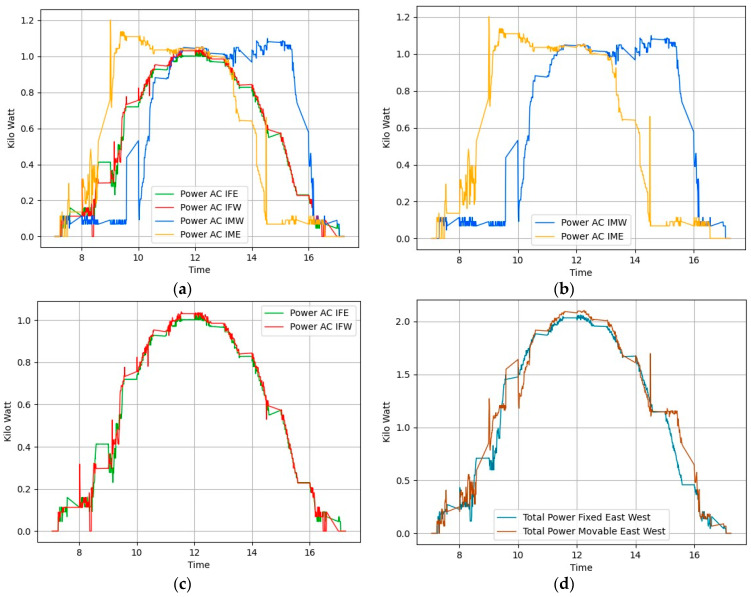
Power generation plots of all solar panels in Algorithm 2 Day2. (**a**) All panels. (**b**) Movable east-facing and movable west-facing panels. (**c**) Fixed east-facing and fixed west-facing panels. (**d**) Total power generated by movable versus total power generated by fixed panels.

**Figure 12 sensors-24-01167-f012:**
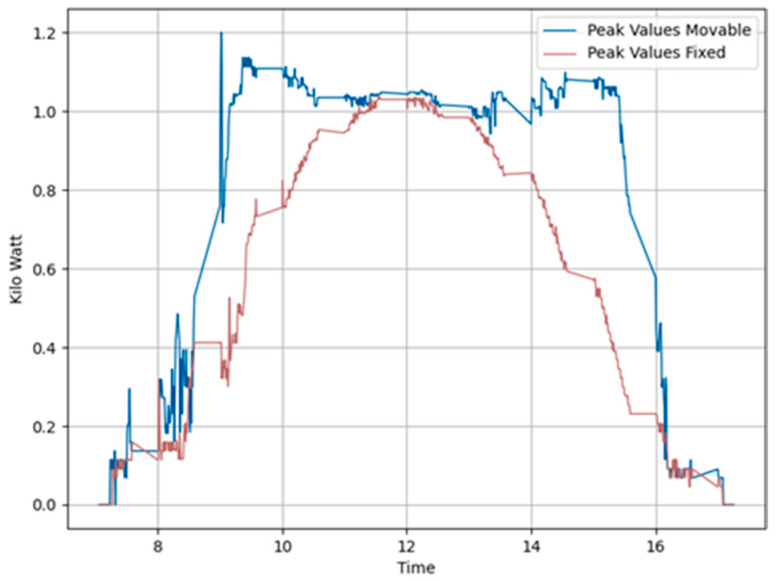
Peak variance ratio of Day 3 (20231121) Algorithm 2.

**Figure 13 sensors-24-01167-f013:**
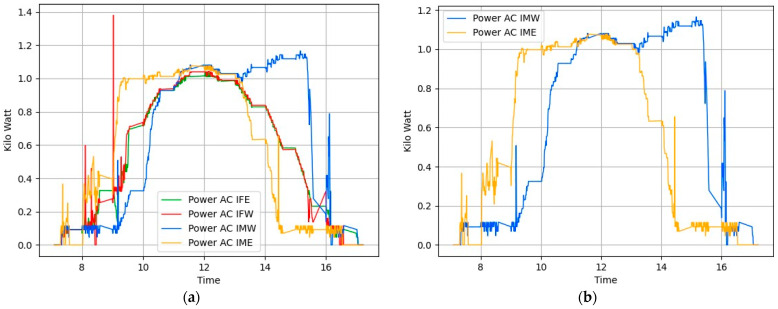
Power generation plots of all solar panels in Algorithm 2 Day 3. (**a**) All panels. (**b**) Movable east-facing and movable west-facing panels. (**c**) Fixed east-facing and fixed west-facing panels. (**d**) Total power generated by movable versus total power generated by fixed panels.

**Figure 14 sensors-24-01167-f014:**
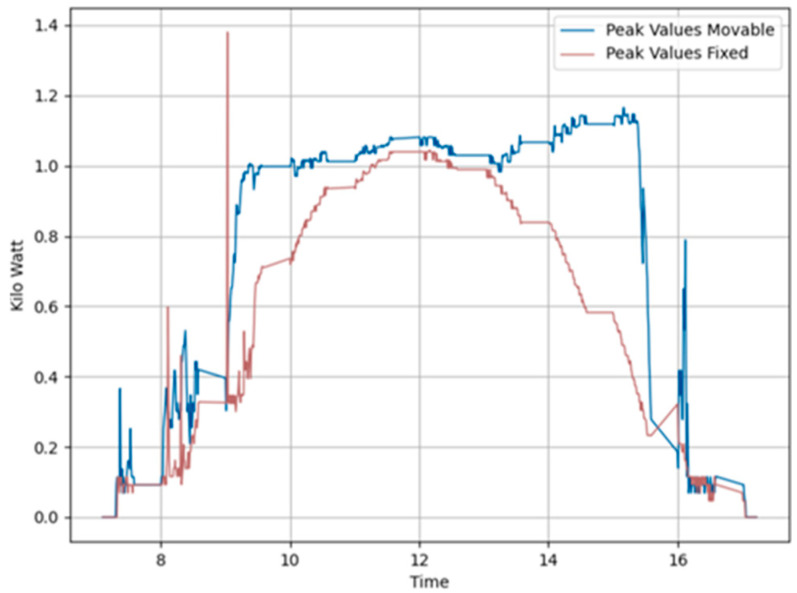
Peak variance ratio of Day 2 (20231121) Algorithm 2.

**Figure 15 sensors-24-01167-f015:**
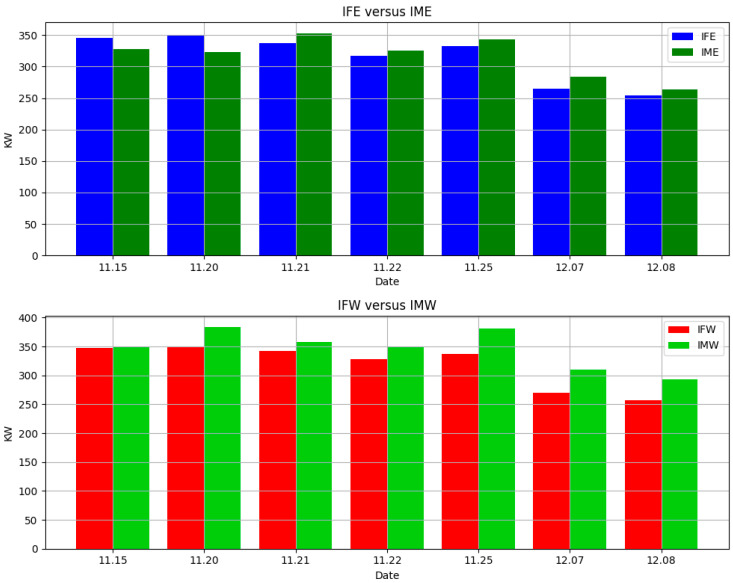
Cumulative power generation comparison of foldable panels versus fixed panels.

**Figure 16 sensors-24-01167-f016:**
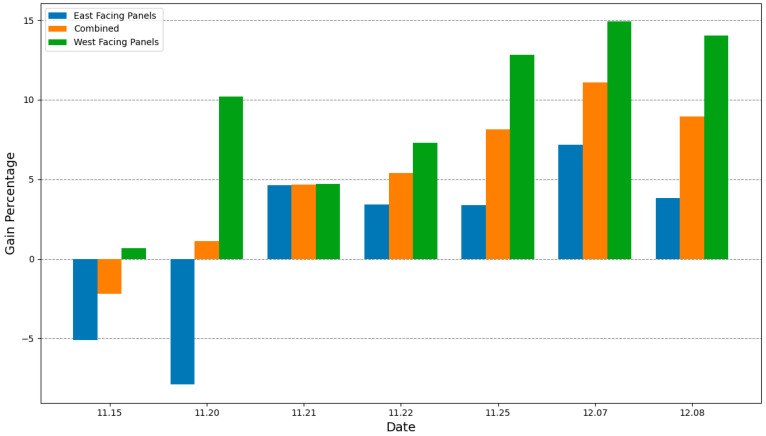
Gain of power generation of foldable panels compared to fixed panels.

**Figure 17 sensors-24-01167-f017:**
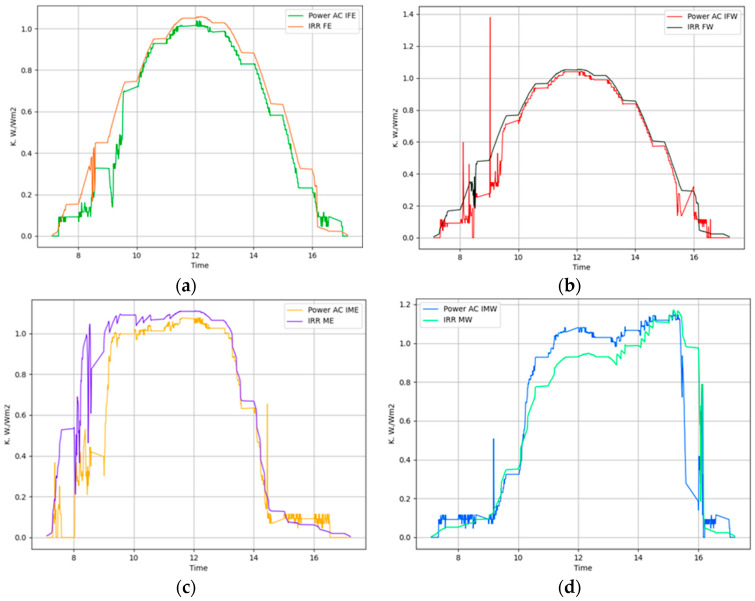
Radiation versus power production in the east-facing panel. (**a**) Fixed east panel; (**b**) fixed west panel; (**c**) movable east panel; (**d**) movable west panel.

**Figure 18 sensors-24-01167-f018:**
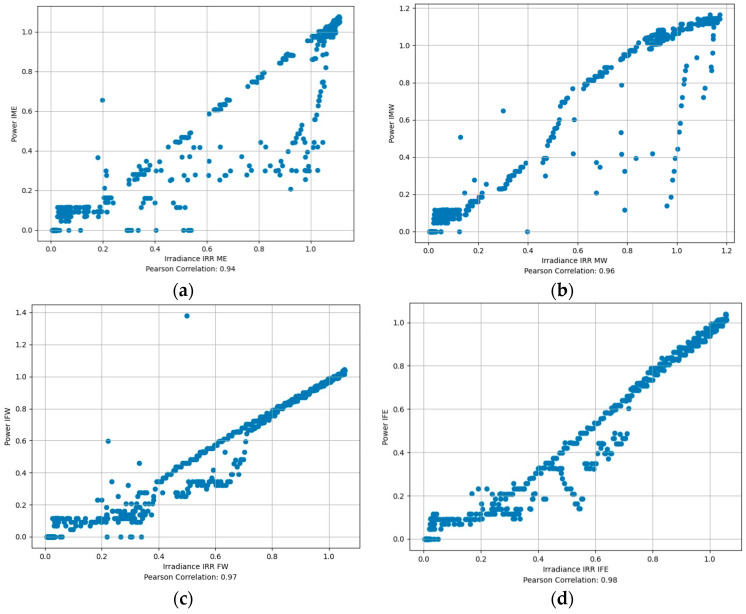
Correlation plot of solar irradiation and power generation. (**a**) Movable east panel, (**b**) Movable west panel (**c**) Fixed west panel (**d**) Fixed east panel.

**Table 1 sensors-24-01167-t001:** Cumulative power generation of fixed and foldable panels, peak variance ratio, and percentage gain in foldable panels.

	P_IFE_ (Kilo-Watt)	P_IFW_ (Kilo-Watt)	P_IME_ (Kilo-Watt)	P_IMW_ (Kilo-Watt)	Peak Variance Ratio (*STD_PP_*)	*PG* (%)
Time-based Solar Tracking (Algorithm 1) [31]	345.98	347.39	328.29	349.73	1.11	−2.21%
Improved Time-based Solar Tracking (Algorithm 2)	336.8	342.37	352.43	358.41	1.13	4.66%
Improved Time-based Solar Tracking (Algorithm 2)	331.92	337.62	343.17	380.90	1.11	8.14%

## Data Availability

The code and dataset shall be made available upon request via email to the corresponding author.

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
