# Peer review of "Design and Performance Analysis of Foldable Solar Panel for Agrivoltaics System"

_sensors, 2024, doi:10.3390/s24041167_

Round 1

Reviewer 1 Report

Comments and Suggestions for Authors

This paper investigates the integration of a foldable solar panel system with a dynamic tracking algorithm for Agrivoltaics Systems (AVS). The results show promise for smart farming by creating microclimates for diverse crops, reducing water evaporation, and minimizing the carbon footprint. This represents a positive step toward sustainable land use and improved food security. The manuscript can be improved if the authors can revise the following points:

1.      The Materials and Methods lack clarity.

2.      The authors claimed that "innovative configuration enables the adjustment of the panel's tilt angle to optimize solar energy capture…." I highly suggest that the author provide a more precise description, supported by references.

3.      The introduction section lacks sufficient information about previous research, particularly in the field of Foldable Solar Panel.

4.      The equation description and the source of this content need to be articulated clearly. I strongly suggest to the author that Table 1 should be thoroughly cross-checked with appropriate references for accuracy.

5.      The authors also asserted, "The peak variance improvement ratio is 1.11, greater than 1, indicating uniform power generation throughout the day …." Please verify the accuracy of this statement.

6.      The figures need significant improvement in quality, especially Figure 9 and 11.

7.      The abstract and conclusion should be synchronized. It requires a substantial correction before publication.

8.      The paper requires a thorough revision by the authors to enhance its overall algorithm and clarity of statements.

9.      Please raise the readability of this manuscript.

Comments on the Quality of English Language

 Minor editing of the English language is required.

Author Response

Thank you for your review and for giving helpful comments which were very useful in revising the paper.

Reviewer 2 Report

Comments and Suggestions for Authors

I feel authors can work on similar works that has been suggested here  https://doi.org/10.3390/su141911880

the authors should introduce role of solar COP26 it is recommended to cite to the related article.

Authors should present the environment and economic analysis.

Author Response

(The authors gave the same response as above.)

Reviewer 3 Report

Comments and Suggestions for Authors

The reviewer's comments on a manuscript submitted to MDPI Sensors 

Title of the paper: "Design and Performance Analysis of Foldable Solar Panel for Agrivoltaics System"

The manuscript studied the design and implementation of a foldable solar panel system (as opposed to fixed panels) with a dynamic tracking algorithm for Agrivoltaics systems applications. The paper is interesting, however it has some drawbacks which must be addressed before it can be considered further. Here is a list of some comments: 

The English writing of the paper must be substantially improved. There are many typos and grammatical errors which must be fixed.

The novelty of the work is not clear. What's different in this foldable solar system as compared to similar ones? How does it work differently? What is unique in this system? What advantages does it offer? 

There are some published studies which are missing in the Introduction Section. 

The authors claim that their system incorporates a solar tracking mechanism adjusting the panels' tilt angles to ensure they are optimally positioned for maximum energy generation. There should be some evidences on how the energy production will be maximised under such mechanism. 

Is this the first time such AV system is proposed? If yes, how did the authors come up with such design configuration? 

Further explanation is required on how the tracking algorithm works and shadow effects were measured. 

It wasn't cleat what the inputs and outputs of the algorithm were. It will be useful to explicitly list them.  

The experiment section is probably the most interesting part of the paper. However, it lacks a comprehensive discussion of the results. The current discussion doesn't provide useful information about how efficient the design configuration was, how accurate the algorithm worked, and how valid the results will be for a real-scale experiment. The results are presented graphically, that is nice, but some figures must be explained further, e.g. Figure 7-11. 

Conclusion is too short. It should be little expanded by including a summary of findings and some directions for future research.  

Comments on the Quality of English Language

Extensive editing of English language required

Author Response

(The authors gave the same response as above.)

Round 2

Reviewer 1 Report

Comments and Suggestions for Authors

I suggest that the authors can further refine it carefully before the paper is accepted.

Comments on the Quality of English Language

Minor editing of the English language is required.

Reviewer 2 Report

Comments and Suggestions for Authors

accept

Reviewer 3 Report

Comments and Suggestions for Authors

The revised version has been substantially improved and the authors addressed the comments to much extent. I would like to suggest to accept the revised paper in its current form. 

Comments on the Quality of English Language

It will be useful if authors can do a proofread of the paper before it's going to be published.